# Subjective burden of government-imposed Covid-19 restrictions in Switzerland: Evidence from the 2022 LINK Covid-19 survey

**Günther Fink**[1,2]*, **Katharina Förtsch**[3], **Stefan Felder**[4]

**1** University of Basel, Basel, Switzerland, **2** Swiss Tropical and Public Health Institute, Allschwil, Switzerland, **3** LINK, Zürich, Switzerland, **4** Department of Economics, University of Basel, Basel, Switzerland

* guenther.fink@swisstph.ch

## Abstract

### Background

While a large literature has quantified the health and economic impact of COVID-19, estimates on the subjective losses in quality of life due to government imposed restrictions remain scarce.

### Methods

We conducted a nationally representative online survey in Switzerland in February 2022 to measure average self-reported quality of life with government restrictions. We used a discrete choice experiment to compute average willingness to pay for avoiding specific restrictions and time-trade-off questions to quantify the relative quality of life under restrictions.

### Results

A total of 1299 Swiss residents completed the online survey between February 9th and 15th, 2022. On average, respondents valued life under severe restrictions at 39% of their usual life (estimated relative utility 0.39 [0.37, 0.42]). Willingness to pay for avoiding restrictions was lowest for masks (CHF 663 [319, 1007]), and highest for schools and daycares (CHF 4123 [3443, 4803]) as well as private parties (CHF 4520 [3811, 5229]). We estimate that between March 2020 and February 2022 a total of 5.7 Million QALYs were lost due to light, moderate and severe restrictions imposed by the governments.

### Conclusions

The quality of life losses due to government restrictions are substantial, particularly when it comes to the closure of schools and daycares, as well as the prohibition of private gatherings. Future policies should weigh these costs against the health benefits achievable with specific measures.

**Data Availability Statement:** All data is available in the Supporting Information.

**Funding:** The author received no specific funding for this work

**Competing interests:** The authors have declared that no competing interests exist.

## Introduction

The Covid-19 pandemic has affected the global community like few epidemics before; as of June 15[th], 2022, more than half a billion cases have been documented, and 6.3 million individuals have died [1]. To prevent even larger adverse health impacts and to ensure the continued functionality of their health systems, governments around the globe have relied on a range of non-pharmaceutical interventions, ranging from mandatory wearing of masks, school closures and home office requirements, to complete lockdowns [2]. While almost the entire global population has personally experienced these measures and while a large literature has tried to assess their effectiveness [3–6], relatively little is known about the impact of these measures on subjective well-being as well as the overall quality of life.

A large and rapidly growing body of evidence has highlighted specific aspects of wellbeing affected by government measures, such as the loss of early life learning opportunities [7], limited access to schooling [8], loss of employment [9], and social isolation [6]. Recent literature has also highlighted the increased incidence of loneliness [10] and mental health problems among adolescents [11] and adults [12] as well as a general deterioration of living conditions, particularly in low- and middle-income countries [13].

All of these effects are important, but cover only specific aspects of individual well-being, without fully capturing the impact of (not necessarily warranted [14, 15]) government restrictions on the overall quality of life of the population exposed [16]. In previous work [17], we attempted to generate such comprehensive estimates, using data collected from France, India, Italy, the UK and the US through the MTurk online platform. Using a sample of online volunteers recruited through the MTurk platform, we showed that average quality of life losses due to government restrictions were large across all countries [17]. Additional research from Australia [18], France [19] and Spain [20–22] and Sweden [23] suggests that the welfare losses due to government restrictions are substantial, but the majority of the population is willing to accept such measures if they can reduce the risk of health system overload [19] or excess mortality [18].

Relative to many other countries, government restrictions in Switzerland were weaker, with strict lockdown measures and school closures only enacted in the first half of 2020 [24]. Despite these less restrictive policies, resistance to government measures was substantial in Switzerland, with continued protests against government mandates throughout the pandemic [25].

In order to quantify the average subjective utility losses in Switzerland, we embedded previously developed survey modules in an ongoing national survey in Switzerland in February 2022 and present the main findings of this survey here.

## Methods

### Study design

This study was designed as a cross-sectional study using data collected through a nationally representative online survey. To quantify marginal willingness to pay to avoid certain restrictions, we conducted a discrete choice experiment following the guidelines of the International Society for Pharmacoeconomics and Outcomes Research (ISPOR) [26].

To quantify total losses in quality of life, we used data on government restrictions from the Oxford Covid-19 Government Response Tracker (https://doi.org/10.1038/s41562-021-01079-8) [2]. Data on the number of Covid-19 deaths by 10-year age group, sex and canton were downloaded from the Swiss Ministry of Health (BAG) on March 25[th], 2022.

## Survey participants

Survey questions were administered through LINK as part of their ongoing national Covid-19 surveys. LINK specializes in online research in Switzerland and conducts about 500–600 online surveys per year (https://www.link.ch/). LINK uses a nationally representative panel sample of 115,000 adult respondents for all of their surveys–this core sample was recruited nation-wide through computer-assisted telephone interviews. For each survey round, an invitation to participate was sent to all eligible participants in this survey pool. Participation was rewarded with a small gift voucher (with an approximate value of CHF 1 for a short survey) for each successfully completed survey. Given that the relative participation of all groups is directly observable, sampling weights can be created that make the study population representative of the Swiss population with respect to age, gender, region, education and household income.

All surveys were translated to French, German and Italian. All original questions asked are available in Appendix 1. All only surveys were completed between February 9th and 15th, 2022.

## Inclusion / exclusion criteria

All respondents in the LINK panel were included in the study as long as they fulfilled the following requirements: 1) currently living in Switzerland; 2) Using the internet at least once a week for private purposes; 3) able to complete the questionnaire in either French, German or Italian; and 4) age 18–79 years.

## Primary outcome variables

The primary outcome of interest were the subjective losses in subjective quality of life due to Covid-19 specific restrictions. We quantified these losses in two ways. First, and following standard procedures for measuring disease-specific quality of life [27], we asked subjects to answer a set of standardized time-tradeoff questions. The specific questions we asked were:

> *"First, consider a scenario where **you are required to wear a mask in public at all times, are not allowed to go to restaurants, clubs or the gym, and travel is prohibited**. If you were given the choice of living with these limitations and your normal life*:
>
> *. . .would you rather live your normal life for X months (option A) or **12 months** in this kind of strict lockdown (option B)?"*

All subjects started with a choice between 12 months of normal life against 12 months with restrictions. If they preferred 12 months of normal life to life with restrictions (as expected), they were sequentially asked to make a choice between 10 months of normal life vs. 12 months with restrictions, then 8 months of normal life, then 6, 4, 3, 2, 1, and 0 months–all against 12 months of life with restrictions.

We also asked a similar set of questions for even stricter restrictions:

> *"Instead, imagine an even stricter lockdown scenario where **you are required to wear masks in all public spaces, cannot eat, drink, go to clubs or the gym, private parties and events are banned**, **all children must be homeschooled and you are not allowed to travel.** If you had the choice between living in this kind of lockdown and your normal life, would you rather live your normal life for X months (option A) or **12 months** in this kind of strict lockdown (option B)?"*

In previous work, we had also considered a more standard end-of-life framing, where subjects were asked to trade off 10 years of life with restrictions against a smaller amount of years without restrictions [17]. These questions yielded very similar results; given that our pilot participants perceived it to be somewhat difficult to live longer period with restrictions, we opted for the shorter-term framing in this study.

In order to quantify the respondents' willingness to pay for avoiding specific restrictions, we conducted a discrete choice experiment within the same survey. Within this experiment, each survey respondent was asked to choose between bundles of living conditions involving restrictions on everyday life as well as pre-specified monthly incomes. Given that we did not want subjects to trade off the benefits of measures against the perceived costs, we chose a framing that forced subjects to think about the restriction by itself in the context of picking a place to live:

> "*Imagine a world without COVID-19. You must choose to live in one of the following two countries. The countries differ both in the salary you earn and the restrictions that the government has decided on for everyday life. In which of the two countries would you **prefer** to live and work*?"

In our previous study, we directly compared this neutral residential choice farming against a Covid-19 specific framing, and did not find any systematic differences in the responses, suggesting that the exact framing has only limited impact on average response patterns [17].

In both the original and the Swiss study, each subject was (sequentially) presented with 6 randomly selected vignettes, each containing an Option A and Option B characterized by random variations of the attributes outlined in Table 1:

The six attributes were selected from a list of measures captures in the Oxford Tracker, and adjusted slightly based on the feedback obtained in two rounds of previous testing. For all attributes, we only considered simple YES/NO levels.

Net salary levels were chose to correspond approximately to the 25th, 50th and 75th percentiles of the current Swiss income distribution. Appendix 2 shows 3 out of the 24 vignettes used as well as the average choices made for these vignettes.

## Statistical analysis

We first estimated average utility weights by sex, age group and region for the pooled sample. To avoid biases emerging from extreme preferences (subjects stating they preferred a life with restrictions to a life without restrictions or subjects stating they would prefer 0 months of healthy life to 12 months of life with restrictions) we also estimated average utility weights in a restricted sample of subjects with an interior switching point (switching point range 1–10). To derive relative utility we divided the observed willingness to pay (switching point) by 12.

**Table 1. Attribute levels.**

| Attribute | Levels |
| --- | --- |
| Monthly Net Salary: | CHF 5000/6500/8500 |
| No restaurants, bars and clubs | YES/NO |
| No sports facilities for you to exercise | YES/NO |
| Mandatory wearing of masks in public | YES/NO |
| No schools or day care centers (home schooling only) | YES/NO |
| Travel abroad only with official permission | YES/NO |
| No private parties, weddings or concerts allowed | YES/NO |

For the discrete choice experiment, responses were analyzed using a random utility framework [28]. The relative impacts (marginal effects) of each attribute on the choice made were estimated using conditional logistic regression models [29] and compared to (scaled by) the relative weight of income in these decisions. To ensure data quality, we evaluated the proportion of respondents always choosing the first or second option.

Last, we used data on the duration of light and severe Covid-19 restrictions in Switzerland to compute the estimated number of quality-adjusted life years lost. Data on Covid-19 restrictions was taken from the Oxford Covid-19 Government Response Tracker [2] and divided into periods with severe restrictions (stringency index > 70), moderate restrictions (stringency index 50–79) and light restrictions (stringency index 20–49). Daily data on the stringency index in Switzerland are shown in S1 Fig. To quantify the total number of quality-adjusted life years lost, we multiplied the number of days under severe restrictions with the estimated utility weights computed. Given the large heterogeneity in the time tradeoff questions, we considered three scenarios: in our first and preferred scenario, we excluded all extreme preferences from the analysis and used only stated preferences between 1 and 11 months for severe restrictions. For moderate and mild restrictions, we simply interpolated disutilities between the severe case and 1 (equal intervals). In our second scenario, we used all time tradeoff data both for severe and for moderate restrictions. For mild restrictions, we once again used interpolation, taking the average between 1 (no disutility) and the moderate restrictions disutility weight. Last, we also considered a third scenario where we allowed for indifference between life with and without restrictions, but excluded subjects unwilling to trade off any time against life with restrictions. This one-sided data censoring is likely to bias average preferences towards lower disutilities (because we remove subjects who are most averse to restrictions) and should thus provide a lower bound for the true disutility generated by restrictions. S1 Table provides further details on the calculations of QALYs and the disutilities used in the three scenarios.

## Ethical considerations

All surveys were completed anonymously online. All respondents provided written consent to the use of data for research by ticking a box before the questionnaire started. Due to the absence of identifiable data, the study was rated as non-human subjects research by the ethics commission for Northwestern and Central Switzerland (Ethikkommission Nordwest- und Zentralschweiz) in EKNZ Req 2021.00616.

## Results

As shown in Table 2, a total of 1299 respondents completed the online survey between February 9th and 15th, 2022. 49.7% of respondents were female, and 76% of respondents indicated to be currently working; 22.9% were below age 30, and 19.6% between ages 60 and 79.

On average, respondents were willing to give up only about 4 months of their usual life against 12 months of life with severe restrictions (Fig 1; see S2 Fig for light restrictions). In the severe restrictions scenario, 233 subjects (17.9 percent) indicated that they valued life with and without restrictions equally and 626 respondents (48.2%) indicated that they were not willing to give up any of their normal life for 12 months of life with restrictions.

When all responses were considered (Fig 2A), mean utility weights were lowest among females from the German speaking part of Switzerland, with an estimated utility weight of 0.26 (95% CI [0.29, 0.22]). Mean utility weights were highest for among males in the Italian-speaking part with an estimated mean utility weight of 0.49 [0.40, 0.57]. When subjects with extreme preferences (response = 12 or 0) were excluded (Fig 2B), mean utility weights were slightly

**Table 2. Sample characteristics by region.**

| | German-speaking | | French-speaking | | Italian-speaking | | Overall | |
|---|---|---|---|---|---|---|---|---|
| | N = 802 | | N = 275 | | N = 222 | | N = 1299 | |
| Characteristic | N | % | N | % | N | % | N | % |
| Female | 395 | 49.3% | 140 | 50.9% | 110 | 49.5% | 645 | 49.7% |
| Working | 630 | 78.6% | 193 | 70.2% | 167 | 75.2% | 990 | 76.2% |
| Age 15–29 | 178 | 22.2% | 71 | 25.8% | 49 | 22.1% | 298 | 22.9% |
| Age 30–44 | 226 | 28.2% | 79 | 28.7% | 64 | 28.8% | 369 | 28.4% |
| Age 45–59 | 237 | 29.6% | 74 | 26.9% | 67 | 30.2% | 378 | 29.1% |
| Age 60+ | 161 | 20.1% | 51 | 18.5% | 42 | 18.9% | 254 | 19.6% |

higher, with lowest utility for females in the French part (0.35 [0.28, 0.43] and highest utility for males in the Italian part (0.46 [0.37, 0.56]).

Fig 3 shows mean utility weights by age group and sex. Mean utility weights were lowest among females 30–44, with a mean utility of 0.24 [0.19, 0.29], and highest among men 60–79 with a mean utility of 0.36 [0.28, 0.44] in the pooled sample (Panel A). When subjects with extreme preferences were excluded, mean utility increased to an average of 0.39 [0.37, 0.42] (Fig 3, Panel B).

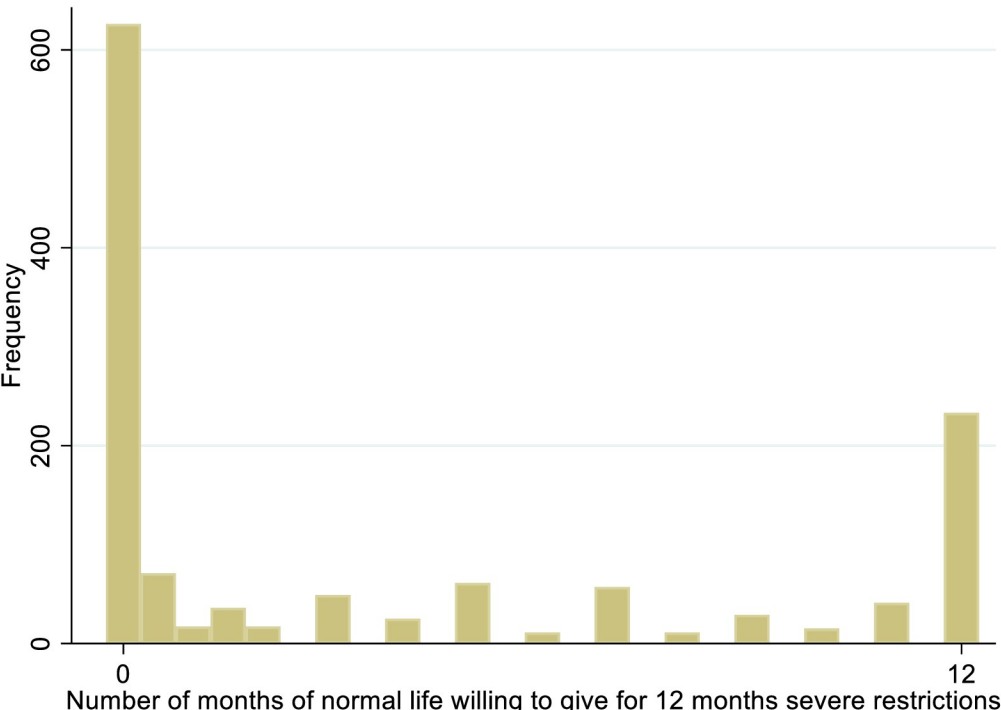

**Fig 1. Time trade-offs: Months of healthy life preferred to 12 months of restricted life.** Notes: Based on the question "*Imagine an even stricter lockdown scenario where you are required to wear masks in all public spaces, cannot eat, drink, go to clubs or the gym, private parties and events are banned, all children must be homeschooled and you are not allowed to travel. Would you rather have x months of your normal life, or 12 months of life with these restrictions?*" Frequencies represent unweighted counts.

 

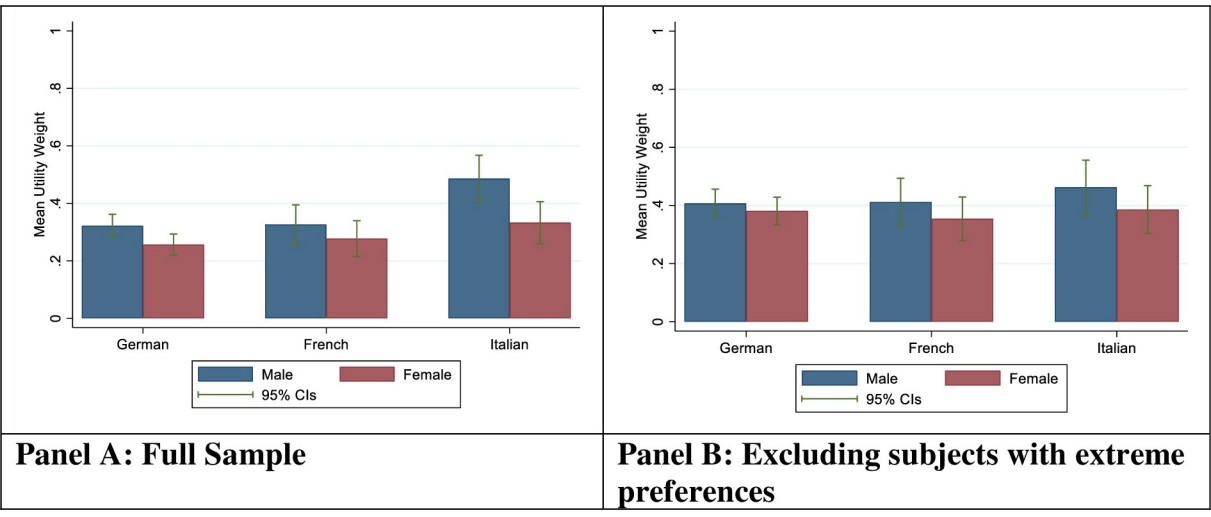

**Fig 2. Average utility weights by sex and region.** Notes: Fig 2 shows estimated average utility weights by language region and sex in the full sample (Panel A) as well as the restricted sample excluding extreme preferences. All utility weights are normalized to range between 0 and 1, with 0 corresponding to zero utility and 1 corresponding to the utility of a fully healthy life.

Table 3 summarizes the main results of the discrete choice experiment. A total of 7794 decisions were recorded and analyzed. All attributes were highly predictive of subjects' choices made. Except for masks, the relative weight given to all restrictions exceeded the weight given to a 1000 CHF salary increase for all subgroups.

Fig 4 summarizes implicit average valuations for the restrictions considered. On average, monthly willingness to pay was lowest for masks with a mean willingness to pay of CHF 663 [319, 1007] and highest for schools and daycares (CHF 4123 [3443, 4803]) as well as private parties (CHF 4520 [3811, 5229]).

Table 4 summarizes the estimated QALY losses. Over the period from January 1 2020 to February 28, 2022, 41 days of strict restrictions (stringency index > 70), 392 days of moderate

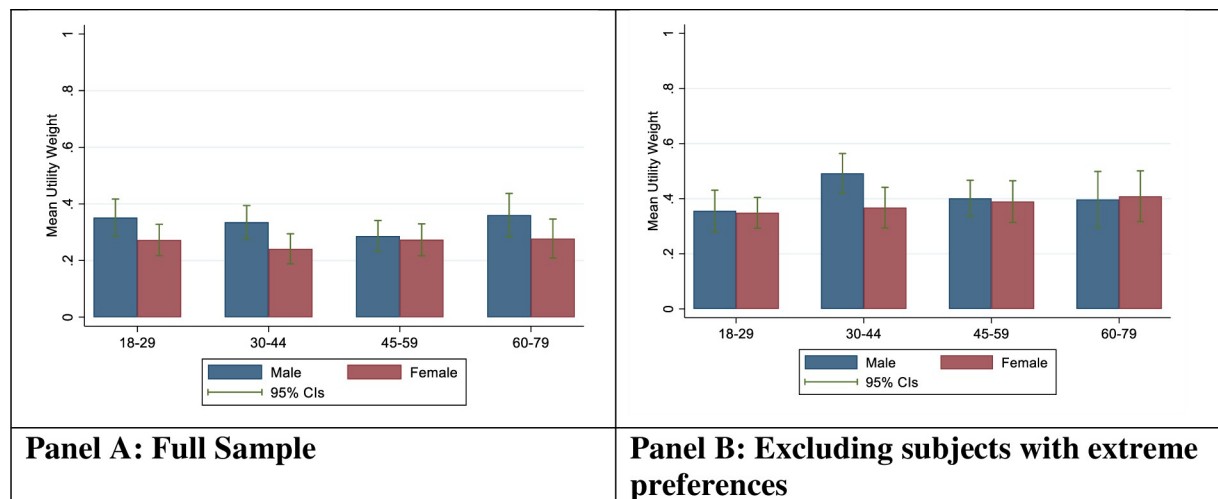

**Fig 3. Average utility weights by age group and gender.** Notes: Fig 2 shows estimated average utility weights by age group and sex in the full sample (Panel A) as well as the restricted sample excluding extreme preferences. All utility weights are normalized to range between 0 and 1, with 0 corresponding to zero utility and 1 corresponding to the utility of a fully healthy life.

**Table 3. Discrete choice experiment: Marginal effects of traits by subsample.**

| Sample | All | German | French | Italian | Female | Male |
|---|---|---|---|---|---|---|
| | (1) | (2) | (3) | (4) | (5) | (6) |
| Net monthly salary 1000s CHF | 0.20*** | 0.18*** | 0.23*** | 0.27*** | 0.12*** | 0.27*** |
| | (0.17, 0.22) | (0.15, 0.21) | (0.17, 0.28) | (0.20, 0.34) | (0.08, 0.16) | (0.23, 0.31) |
| No restaurants, bars and clubs | -0.56*** | -0.56*** | -0.59*** | -0.47*** | -0.51*** | -0.60*** |
| | (-0.63, -0.49) | (-0.64, -0.48) | (-0.72, -0.45) | (-0.64, -0.31) | (-0.61, -0.41) | (-0.70, -0.50) |
| No sports facilities for you to exercise | -0.28*** | -0.32*** | -0.17** | -0.11 | -0.27*** | -0.28*** |
| | (-0.34, -0.21) | (-0.40, -0.24) | (-0.30, -0.04) | (-0.26,- 0.03) | (-0.36, -0.17) | (-0.38, -0.19) |
| Mandatory wearing of masks in public | -0.13*** | -0.15*** | -0.06 | -0.04 | -0.11** | -0.15*** |
| | (-0.19, -0.07) | (-0.23, -0.07) | (-0.18, 0.06) | (-0.15, 0.07) | (-0.20, -0.02) | (-0.24, -0.06) |
| No schools or day care cen-ters (home schooling only) | -0.81*** | -0.82*** | -0.73*** | -0.88*** | -0.91*** | -0.72*** |
| | (-0.88, -0.73) | (-0.92, -0.73) | (-0.90, -0.56) | (-1.07, -0.69) | (-1.02, -0.79) | (-0.83, -0.62) |
| Travel abroad only with state approval | -0.42*** | -0.48*** | -0.21*** | -0.41*** | -0.46*** | -0.39*** |
| | (-0.48, -0.35) | (-0.57, -0.40) | (-0.34, -0.09) | (-0.54, -0.27) | (-0.56, -0.36) | (-0.47, -0.30) |
| No private parties, wed-dings or con-certs allowed | -0.88*** | -0.87*** | -1.01*** | -0.55*** | -0.96*** | -0.83*** |
| | (-0.96, -0.81) | (-0.96, -0.78) | (-1.18, -0.84) | (-0.72, -0.38) | (-1.08, -0.84) | (-0.93, -0.72) |
| N (decisions) | 7794 | 9624 | 3300 | 2664 | 7740 | 7848 |

Notes: All coefficients based on conditional logistic regression model with decision fixed effects. Coefficients represent logit differences; 95% confidence intervals in parentheses. Standard errors are clustered at the individual level, with six responses by subject. All regressions are weighted to achieve nationally representative sample ages 15–79.

\*\*\* $p < 0.01$

\*\* $p < 0.05$

\* $p < 0.1$

restrictions (index 50–70) and 295 days of mild restrictions (20–49) were recorded. Applying the utility weights based on non-extreme preferences (Scenario 1) implies a total QALY loss of 5.7 million. When we based our estimates on all reported preferences (Scenario 2), the estimated impact increased to 9.1 Million QALYs. When we allowed indifferent subjects but removed subjects not willing to give up any normal life for life with restrictions, the estimated impact was 3.8 Million QALYs.

## Discussion

This paper reports the results of the first nationally representative estimates of the quality of life losses generated by government-imposed restrictions in Switzerland between March 2020 and February 2022. The results presented suggest that on average Swiss respondents consider their quality of life under government restrictions to be rather low. When asked to make choices between longer periods of life with restrictions against shorter periods without restrictions subjects were on average willing to give up only about 4 months of their usual life against 12 months of severe restrictions. This implies that severe restrictions reduce the average quality of life by more than 50%. Using this reduction in standard decision science framework implies that even short periods of strict lockdowns–such as the first period in the spring of 2020 –result in a rather massive loss of quality-adjusted life years. Estimating the extent to which these losses are due to government measures vs. the pandemic itself is difficult; if the entire population had opted for these measures without government intervention, the utility losses reported here would be primarily due to the pandemic, and the net impact of government restrictions would have been zero. On the other hand, if nobody had opted for these measures in the

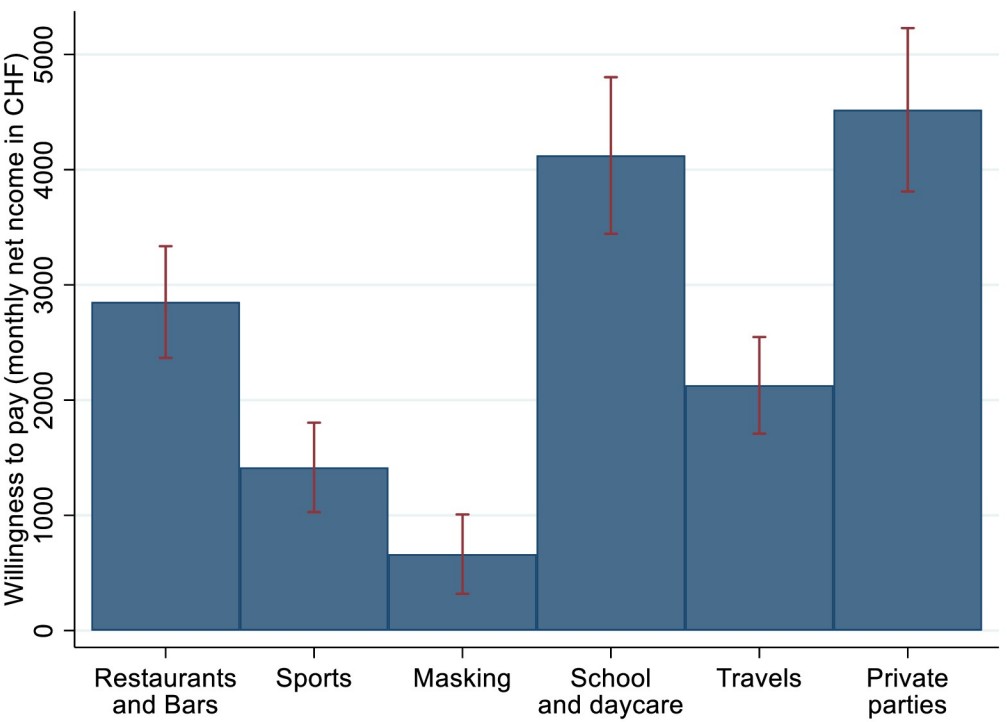

**Fig 4. Implicit willingness to pay for avoiding restrictions.** *Notes*: Estimates based on non-linear combination of point estimates reported in Table 2. Blue bars represent mean valuation, red lines 95% confidence intervals.

absence of government legislation, most of the observed disutility would be due to legislation rather than the pandemic itself.

The rather large losses in the subjective quality of life are also clearly visible in respondents' implicit willingness to pay to avoid specific restrictions on their life. On average, respondents indicated to be willing to give up a bit more than CHF 600 per month for not having to wear masks, and more than CHF 4000 per month for not being allowed to have private parties or for not having to teach children at their homes. These estimates seem large, especially for individuals with income below the median and should be interpreted with caution, as it is not clear whether all subjects would really make these choices when faced with them in reality. Nevertheless, the estimates presented here suggest that most respondents would be willing to give up a substantial share of their income to avoid restrictions, and that restrictions on schooling, private parties and on going out are particularly undesired by the Swiss population.

The overall disutility from life under restrictions is remarkably large. Under a rather pessimistic assumption that the pandemic would have resulted in the deaths of 1% of the total population the total number of life years lost would have been around 650,000. Our most conservative estimate for the total utility loss is close to 4 million QALYs.

The analysis presented here has several other limitations worth highlighting. First, and most importantly, all questions asked were hypothetical, which raises concerns that subjects may overstate their willingness to pay for removing restrictions [26]. Second, it also seems possible that some subjects did not fully understand some of the questions asked in the survey; this may be particularly relevant for the time-tradeoffs, where a surprisingly large proportion of subjects either indicated to prefer restrictions to normal life, or stated that they would rather

**Table 4. Estimated QALY losses due to Covid-19 related government restrictions from January 2020 to February 2022.**

| Canton | Population | Estimated QALY Losses | | |
|---|---|---|---|---|
| | | Scenario 1 | Scenario 2 | Scenario 3 |
| Aargau | 678'207 | 452'700 | 723'127 | 302'965 |
| Appenzell Innerrhoden | 16'145 | 10'777 | 17'214 | 7'212 |
| Appenzell Ausserrhoden | 55'234 | 36'868 | 58'892 | 24'674 |
| Bern | 1'034'977 | 690'842 | 1'103'527 | 462'339 |
| Basel Landschaft | 288'132 | 192'327 | 307'216 | 128'713 |
| Basel Stadt | 194'766 | 130'005 | 207'666 | 87'005 |
| Fribourg | 318'714 | 212'740 | 339'824 | 142'374 |
| Genève | 499'480 | 333'401 | 532'562 | 223'125 |
| Glarus | 40'403 | 26'969 | 43'079 | 18'049 |
| Graubünden | 198'379 | 132'417 | 211'518 | 88'619 |
| Jura | 73'419 | 49'007 | 78'282 | 32'797 |
| Luzern | 409'557 | 273'377 | 436'683 | 182'955 |
| Neuchâtel | 176'850 | 118'047 | 188'563 | 79'001 |
| Nidwalden | 43'223 | 28'851 | 46'086 | 19'308 |
| Obwalden | 37'841 | 25'259 | 40'347 | 16'904 |
| Sankt Gallen | 507'697 | 338'885 | 541'324 | 226'796 |
| Schaffhausen | 81'991 | 54'729 | 87'422 | 36'627 |
| Solothurn | 273'194 | 182'356 | 291'289 | 122'040 |
| Schwyz | 159'165 | 106'242 | 169'707 | 71'101 |
| Thurgau | 276'472 | 184'544 | 294'784 | 123'504 |
| Ticino | 353'343 | 235'855 | 376'746 | 157'844 |
| Uri | 36'433 | 24'319 | 38'846 | 16'275 |
| Vaud | 799'145 | 533'425 | 852'075 | 356'990 |
| Valais | 343'955 | 229'588 | 366'736 | 153'650 |
| Zug | 126'837 | 84'663 | 135'238 | 56'660 |
| Zürich | 1'520'968 | 1'015'239 | 1'621'707 | 679'439 |
| **Switzerland** | **8'544'527** | **5'703'431** | **9'110'462** | **3'816'966** |

Notes: Table 4 shows current population as well as estimated QALY loss due to light (295 days), moderate (392 days) and severe (41) days by canton and for Switzerland overall. Estimates shown in scenario 1 are based on utility weights leaving out extreme preferences; estimates shown in scenario 2 are based on utility weights using all stated preferences, and estimates in scenario 3 are based on utility weights leaving out only subjects reporting to not be willing to give up any of their normal life for life with restrictions.

not have any life at all rather than life with restrictions. It is possible that some subjects indicated that they would rather die than accept measures simply wanted to express their dissatisfaction with government restrictions; it also seems plausible that some respondents expressed indifference simply because they felt they had no choice in any case. Even when these "non-voting" respondents with extreme preferences were excluded, estimated disutilities from living with restrictions changed only marginally, and remained very high compared to the previous international study [17]. Overall, these results suggest that the Swiss population feels more strongly restricted by government containment measures, which would certainly be consistent with the generally more lenient Swiss containment policies compared to neighboring countries. It also seems plausible that responses could change with different framing: our time-tradeoff questions focused on a 12-month horizon, and it possible–even if not obvious–that more higher utility weights would emerge with longer term or end-of-life questions. Similarly, our discrete choice experiment focused on a neutral setting, where subjects had to trade off life

with restrictions against salary in the absence of Covid-19. While this framing was intentionally chosen to prevent subjects from trading off potential disease benefits against the disutility from these measures, answers could be different if shorter-term and disease-specific measures would be considered. In our previous studies, we randomized the framing and found that this did not make much of a difference [17]. Our model also did not incorporate differences in restrictions across cantons–these deviations from national policies were on average fairly minor and likely would not change any of our main results.

Despite these limitations, the main message emerging from this study is clear: government restrictions to contain the spread of infectious diseases cause major losses in the people's quality of life. In the Swiss context, these losses appear particularly large for the prohibition of private meetings and get-togethers as well as for the closure of schools and daycare, and relatively minor for the wearing of masks in public. These private costs associated with each containment measure should be weighed against potential disease transmission benefits in future policy decisions.

## Supporting information

**S1 Appendix. Original questions in French, German and Italian.**
(DOCX)

**S2 Appendix. DCE examples.**
(DOCX)

**S1 Fig. Oxford stringency index Switzerland.**
(DOCX)

**S2 Fig. Time-tradeoffs light restrictions.**
(DOCX)

**S1 Table. Modeling details for QALY calculations.**
(DOCX)

**S1 Data.**
(ZIP)

## Author Contributions

**Conceptualization:** Günther Fink.

**Data curation:** Günther Fink, Katharina Förtsch.

**Formal analysis:** Günther Fink.

**Investigation:** Günther Fink, Katharina Förtsch, Stefan Felder.

**Methodology:** Günther Fink, Stefan Felder.

**Software:** Katharina Förtsch.

**Visualization:** Günther Fink.

**Writing – original draft:** Günther Fink, Stefan Felder.

**Writing – review & editing:** Günther Fink, Stefan Felder.

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
