## [Decision Letter · Decision Letter 0]

22 Nov 2022

PONE-D-22-28984Subjective Burden of Government-imposed Covid-19 Restrictions in Switzerland:

Evidence from the 2022 LINK Covid-19 SurveyPLOS ONE

Dear Dr. Fink,

Thank you for submitting your manuscript to PLOS ONE. After careful consideration, we feel that it has merit but does not fully meet PLOS ONE’s publication criteria as it currently stands. Therefore, we invite you to submit a revised version of the manuscript that addresses the points raised during the review process.

We look forward to receiving your revised manuscript.

Kind regards,

Florian Follert

Academic Editor

PLOS ONE

Journal Requirements:

4. Please amend the manuscript submission data (via Edit Submission) to include author Katharina Förtsch.

Reviewers' comments:

Reviewer's Responses to Questions

**Comments to the Author**

1. Is the manuscript technically sound, and do the data support the conclusions?

Reviewer #1: Yes

Reviewer #2: Partly

2. Has the statistical analysis been performed appropriately and rigorously? 

Reviewer #1: Yes

Reviewer #2: Yes

3. Have the authors made all data underlying the findings in their manuscript fully available?

Reviewer #1: Yes

Reviewer #2: Yes

4. Is the manuscript presented in an intelligible fashion and written in standard English?

Reviewer #1: Yes

Reviewer #2: Yes

5. Review Comments to the Author

Reviewer #1: Dear authors,

Good job and I hope you can see publised your paper as soon as possible. If you need more comparative references, please find attached: https://doi.org/10.1111/beer.12431; doi: 10.3389/fpubh.2022.801525; https://doi.org/10.3390/ijerph182412907;
https://doi.org/10.3390/ijerph18041376;
https://doi.org/10.37467/gka-revvisual.v8.2805;
https://doi.org/10.3390/su13094655.

Best regards, Antonio.

Reviewer #2: Overall assessment:

The authors address the negative effects of Covid-restrictions on quality-adjusted life years lost and make an attempt to quantify the restrictions in monetary terms. The topic is highly relevant and I have a lot of a lot of sympathy for their research. However, I have some concerns regarding the methodology, which I perceive as crucial issues. Moreover, I have a number of further points, which the authors should address (in particular, I am very critical of the comparison of quality-adjusted life years and life years lost). Generally, the paper is written well, but there are some inconsistencies and imprecisions that have been confusing. I suggest the authors to carefully check for these.

Crucial issues:

1. Question regarding stricter restrictions (line 88 ff. and 166 ff.):

a. What was the original formulation of the question in German/French/Italian?

First, the last part of the question in line 84/85 is different from line 167/168. In particular, line 167/168 read “Would you rather have x YEARS of your normal life, or 12 MONTHS with these restrictions”. This is very confusing to me.

Second, even without this year/month issue, the question on the time trade-off is in my eyes not very intuitive to answer - I tested this question with several people who got confused.

Thrid, I wonder if some participants misunderstood “cannot eat, drink, go to clubs or the gym” as “cannot eat, drink”, especially as the previous question reads “go to restaurants”, and later questions also mention “bars and restaurants” explicitly. This might seem unlikely in most cases, but it might have been that some subjects misinterpreted the question in this way, which would explain the extremely high fraction of subjects not willing to give up any month of their normal life for 12 months under restrictions.

The authors should provide the original questions in an appendix and take care of a proper and consistent translation in the manuscript. A lot rides on the question (the following results build on it) and potential misunderstandings might undermine the results. As things stand, I am not completely convinced by the measurement.

b. Even if my concerns regarding the formulation of the question is resolved, there are two other concerns. First, the authors correctly note that the scenario is hypothetical and in fact, most people are probably not used to make such a decision. Hence, I am a bit hesitant to interpret the number of months at face value (especially in comparison to actual months of life lost). Second, I wonder about the expressive nature of the question. Those subjects who strongly oppose the restrictions might understate the number of months they are willing to trade-in – simply to express how much they dislike the restrictions. In contrast, those who are willing to trade-in 12 months of normal life for 12 months under restrictions might argue they do not have the choice anyway. They might see the pandemic and the governmental restrictions as unavoidable and take events as they are happening, and/or might value living in general.

For these reasons, I think a (implicit) comparison of “actual estimated life years lost” and “estimated quality-adjusted life years lost due to restrictions” such as in Table 3 is problematic. It is of course important to quantify the costs of restrictions (government and self-imposed) and to point out that these have to be taken into account – I would stop here, however, and do not compare life years lost that are based on very different concepts.

2. Robustness tests:

I would ask the authors to provide two more robustness tests.

First, the authors should provide a similar figure to Figure 1 for the very first question on the time trade-off for the less strict lockdown measure. This would help to judge on whether there is a problem with the understanding of the question on the stricter lockdown. Moreover, the authors should then show the robustness of their results by using the less strict lockdown question (which corresponds most likely to intermediate restrictions) to compute utility weights, and then interpolate for strict (and light) restrictions as done before with utility weights of strict restrictions.

Second, what would happen to the results if only those who are not willing to give any month of their normal life for 12 months under restrictions are excluded? As outlined above, these could be subjects who misunderstood the question. I don’t think the same argument applies to those who do not “value” a month under restrictions less than under normal conditions. Therefore, just focusing on those with an interior switching point to show robustness of results did not convince me – it rules out “extreme preference” (symmetric) but not a lack of understanding (asymmetric). The proposed robustness test would be a more conservative estimate of the effect size than the exclusion of extreme preferences in general.

Major issues:

3. General:

The line between what is a binding governmental restriction and what is a restriction by the pandemic itself is a bit blurry. For example, even in absence of a governmental restriction, I might voluntarily wear a mask. This does not imply that I don’t have a willingness to pay to pay not to wear it (i.e. to live in a world without Covid-19), but the personal cost-benefit evaluation. In this sense, I would argue that it is (inseparable) the pandemic and governmental restrictions, and not only governmental restrictions (as concluded in line 233/234).

4. Framing:

The discrete choice makes an effort to use a neutral setting, which would protect against my last point. However, I am doubtful that simply stating the introductory sentence “Imagine a world without COVID-19” (line 98) is strong enough to “force subjects to think about the restriction by itself” (line 96). First, the majority of subjects will have never experienced such restrictions before Covid, and the intensity of restrictions and the short time lag to the removal of restrictions will potentially tight them strongly to Covid. Second, the previous questions on the time trade-off lead to an implicit Covid-framing. Third, some of the elicited restrictions are clearly possible in a world without Covid (such as travel restrictions in totalitarian regimes), others like “wearing masks in public” seem highly unlikely in a world without infectious disease. Therefore, I am not surprised that there is no difference to previous study with Covid-framing. In summary, I would be more careful in claiming that it is only the governmental restrictions, but would rather see results as the sum of governmental and Covid-imposed restrictions.

5. Discussion:

I strongly disagree with the interpretation and comparison in line 236 to 241. First, I don’t think estimated quality-adjusted life years lost due to restrictions are comparable to life years lost due to higher mortality in an uncontained pandemic – see also my first point. Second, even if we would accept such a comparison, the death of 1% of the population would have severe consequences on the rest of the population. In this case, the benchmark is not the “normal life” anymore. Instead, we would have to do a similar choice experiment as for the restrictions, namely “normal life with functioning health sector” versus “life with collapse of health sector” or “life with old relatives“ versus “life with death of old relatives”. I don’t think it is far-fetched to assume that most people would not change 12 months of normal life against 12 months under such conditions.

This is a rather strong referee request, but I would ask the authors not to make this comparison between actual life year lost and quality adjusted-life years lost in the discussion and also not in Table 3 (see point 1). The comparison is simply not valid for methodological reasons and I think the paper makes an interesting contibution even without the comparison.

6. Willingness to pay:

Summing-up the average willingness to pay, subjects would be willing to spend on average a net monthly income of CHF 16,000 to avoid the restrictions. This amount is outside the monthly budget constraint of most individuals and seems highly inflated. The reason might be similar as before. Subjects who oppose the restriction might overstate their willingness to pay in order to express their strong opposition. The authors point the potentially overstated willingness to pay out in line 255 of the discussion but remain silent on potential reasons and implications.

My suggestion here would be to add that the elicited willingness to pay should not be taken at face value, and potentially to not state the numbers prominently in the abstract. Instead, I would recommend a relative interpretation of restrictions in comparison to the most (or least) “expensive” restrictions (e.g. on average, subjects are willing to pay 6.8-times as much to be allowed to do private parties than to not have to wear masks in public).

Minor issues:

7. Explanation of approach: Complementary to the description how the authors calculated the quality-adjusted life years with the utility weights in the text, I think stating the formal estimation approach would ease the understanding (potentially accompanied by an example). This might be also information for an additional appendix.

8. Typos

- line 89: should it read “live” instead of “leave”?

- line 93: missing space “was asked”

- line 106: “each subject”

- line 119 to line 123: it should read “months” instead of “years”? (questions before are referring to “months”)

- line 163: it should read “normal life” instead of “health life”?

- line 172: it should be either “for” or “among”?

- line 198 (Table 2): some confidence intervals are not displayed correctly (minus sign and brackets covered)

6. PLOS authors have the option to publish the peer review history of their article (what does this mean?). If published, this will include your full peer review and any attached files.

Reviewer #1: **Yes: **Antonio SANCHEZ-BAYON, Applied Economics, Universidad Rey Juan Carlos

Reviewer #2: No

---

## [Author Response · Author response to Decision Letter 0]

17 Feb 2023

Editor Comments

We have formatted the manuscript using PLOS ONE style requirements.

We have updated the “ethical considerations” section in the manuscript, where we now write 

“All surveys were completed anonymously online. All respondents provided written consent to the use of data for research by ticking a box before the questionnaire started. Due to the absence of identifiable data, the study was rated as non-human subjects research by the ethics commission for Northwestern and Central Switzerland (Ethikkommission Nordwest- und Zentralschweiz) in EKNZ Req 2021.00616.”

We have added this – please see our reply to item 2).

4. Please amend the manuscript submission data (via Edit Submission) to include author Katharina Förtsch.

Thank you for flagging this – we have added Katharina to authors in the online system.

 

Reviewer #1 Comments

Dear authors,

Good job and I hope you can see published your paper as soon as possible. If you need more comparative references, please find attached: https://doi.org/10.1111/beer.12431; doi: 10.3389/fpubh.2022.801525; https://doi.org/10.3390/ijerph182412907;
https://doi.org/10.3390/ijerph18041376;
https://doi.org/10.37467/gka-revvisual.v8.2805;
https://doi.org/10.3390/su13094655.

Best regards, Antonio.

Dear Antonio, thanks a million for the kind review and references – we have added all of them to the revised manuscript.

 

Reviewer #2 Comments

 Overall assessment:

The authors address the negative effects of Covid-restrictions on quality-adjusted life years lost and make an attempt to quantify the restrictions in monetary terms. The topic is highly relevant and I have a lot of a lot of sympathy for their research. However, I have some concerns regarding the methodology, which I perceive as crucial issues. Moreover, I have a number of further points, which the authors should address (in particular, I am very critical of the comparison of quality-adjusted life years and life years lost). Generally, the paper is written well, but there are some inconsistencies and imprecisions that have been confusing. I suggest the authors to carefully check for these.

Thank you for the careful review and detailed comments – we really appreciate your detailed comments and helpful suggestions and have done our best to address all comments as outlined in further detail below.

Crucial issues:

1. Question regarding stricter restrictions (line 88 ff. and 166 ff.):

a. What was the original formulation of the question in German/French/Italian?

First, the last part of the question in line 84/85 is different from line 167/168. In particular, line 167/168 read “Would you rather have x YEARS of your normal life, or 12 MONTHS with these restrictions”. This is very confusing to me.

Thank you for catching this – the legend to Figure 1 was incorrect and should have said “months” rather than years – we have fixed this now. As for the original language, we developed the survey simultaneously for all three main Swiss language, i.e., French, German and Italian.

For this specific questions, the three language versions read as follows:

« ..préféreriez-vous avoir <X> mois de votre vie habituelle (option A) ou 12 mois dans ce type d'enfermement strict (option B) ? »

«...würden Sie lieber <X > Monate lang Ihr normales Leben führen (Option A) oder 12 Monate lang in dieser Art von strengem Lockdown (Option B)?»

 « ..preferiresti avere <X> dei mesi di vita senza restrizioni (opzione A) o 12 dei mesi di vita con queste particolare restrizioni (Opzione B)? »

Second, even without this year/month issue, the question on the time trade-off is in my eyes not very intuitive to answer - I tested this question with several people who got confused.

We agree that these hypothetical tradeoffs are not always easy, but piloted these extensively – first in our group (N=30), then in a pilot online (N=50), and then in a relatively large study online (N=950) – overall this seems to work okay even though we agree that these tradeoffs may not be immediately obvious to everyone.

Third, I wonder if some participants misunderstood “cannot eat, drink, go to clubs or the gym” as “cannot eat, drink”, especially as the previous question reads “go to restaurants”, and later questions also mention “bars and restaurants” explicitly. This might seem unlikely in most cases, but it might have been that some subjects misinterpreted the question in this way, which would explain the extremely high fraction of subjects not willing to give up any month of their normal life for 12 months under restrictions.

That is a good question and not easy to answer ex-post. In terms of the exact formulations, all three study languages are slightly more precise than English, directly linking “eating” to “going out” (“andare in ristorante”, “sortir pour diner” and “ausgehen zum Essen”).

The authors should provide the original questions in an appendix and take care of a proper and consistent translation in the manuscript. A lot rides on the question (the following results build on it) and potential misunderstandings might undermine the results. As things stand, I am not completely convinced by the measurement.

Thank you for this suggestion. We have added the exact questions used to the appendix (Appendix 2 of the revised manuscript).

b. Even if my concerns regarding the formulation of the question is resolved, there are two other concerns. First, the authors correctly note that the scenario is hypothetical and in fact, most people are probably not used to make such a decision. Hence, I am a bit hesitant to interpret the number of months at face value (especially in comparison to actual months of life lost). Second, I wonder about the expressive nature of the question. Those subjects who strongly oppose the restrictions might understate the number of months they are willing to trade-in – simply to express how much they dislike the restrictions. In contrast, those who are willing to trade-in 12 months of normal life for 12 months under restrictions might argue they do not have the choice anyway. They might see the pandemic and the governmental restrictions as unavoidable and take events as they are happening, and/or might value living in general.

Both scenarios are certainly possible – we now acknowledge this explicitly in the revised Discussion section of the paper, where we write 

“It is possible that some subjects indicated that they would rather die than accept measures simply wanted to express their dissatisfaction with government restrictions; it also seems plausible that some respondents expressed indifference simply because they felt they had no choice in any case. Even when these “non-voting” respondents with extreme preferences were excluded, estimated disutilities from living with restrictions changed only marginally, and remained very high compared to the previous international study.”

For these reasons, I think a (implicit) comparison of “actual estimated life years lost” and “estimated quality-adjusted life years lost due to restrictions” such as in Table 3 is problematic. It is of course important to quantify the costs of restrictions (government and self-imposed) and to point out that these have to be taken into account – I would stop here, however, and do not compare life years lost that are based on very different concepts.

We agree that these measures are conceptually a bit different, and have removed the direct comparison from Table 3 – as described in further detail below, we now show alternative estimates of the total QALY losses instead.

2. Robustness tests:

I would ask the authors to provide two more robustness tests.

First, the authors should provide a similar figure to Figure 1 for the very first question on the time trade-off for the less strict lockdown measure. This would help to judge on whether there is a problem with the understanding of the question on the stricter lockdown. 

Thank you for this suggestion. We have added a new Figure to the appendix, which shows the empirical distribution of time trade-offs with less strict measures. As you can see below, the distribution looks similar, with an average WTP of 4.2 vs. 3.8 in the severe restrictions case. In hindsight, the choice of the light restriction set was likely not ideal, since both scenarios included masking, closures of bars, restaurants and gyms, and as well as travel bans. The severe restriction added closure of schools and restrictions on private parties, which may not be perceived as major change, and likely explains at least partially the relatively small differences in these measures.

Moreover, the authors should then show the robustness of their results by using the less strict lockdown question (which corresponds most likely to intermediate restrictions) to compute utility weights, and then interpolate for strict (and light) restrictions as done before with utility weights of strict restrictions.

Thank you for this suggestion. We have completely revised Table 3, and now show three different estimates as described in the revised Methods section:

“To quantify the total number of quality-adjusted life years lost, we multiplied the number of days under severe restrictions with the estimated utility weights computed. Given the large heterogeneity in the time tradeoff questions, we considered three scenarios: in our first, and preferred scenario, we excluded all extreme preferences from the analysis and used only used stated utilities strictly larger than 0 and smaller than 1 for severe restrictions. For moderate and mild restrictions in scenario 1, we simply interpolated disutilities between the severe case and 1 (equal intervals). In our second scenario, we used average utilities from the time tradeoff questions for both severe and for moderate restrictions. For mild restrictions, we once again used interpolation, taking the average between 1 (no disutility) and the moderate restrictions disutility weight. Last, we also considered a third scenario where we allowed for indifference between life with and without restrictions, but excluded subjects unwilling to trade off any time against life with restrictions. This one-sided data censoring is likely to bias average preferences towards lower disutilities (because we remove subjects who are most averse to restrictions) and should thus provide a lower bound for the true disutility generated by restrictions.”

Second, what would happen to the results if only those who are not willing to give any month of their normal life for 12 months under restrictions are excluded? As outlined above, these could be subjects who misunderstood the question. I don’t think the same argument applies to those who do not “value” a month under restrictions less than under normal conditions. Therefore, just focusing on those with an interior switching point to show robustness of results did not convince me – it rules out “extreme preference” (symmetric) but not a lack of understanding (asymmetric). The proposed robustness test would be a more conservative estimate of the effect size than the exclusion of extreme preferences in general.

Thank you also for this suggestion. Dropping 0 votes and leaving in 12 votes increased average utility substantially from 0.3 to 0.6. While it does not seem obvious to us to select respondents this way (having access to bars, restaurants and schools should on average have a non-zero option value for all respondents, which suggests that a vote of 12 also implies not understanding the question), we believe that this coding can be useful as a lower bound estimate, and have incorporated this as Scenario 3 in the revised Table 3 as described above.

Major issues:

3. General:

The line between what is a binding governmental restriction and what is a restriction by the pandemic itself is a bit blurry. For example, even in absence of a governmental restriction, I might voluntarily wear a mask. This does not imply that I don’t have a willingness to pay to pay not to wear it (i.e. to live in a world without Covid-19), but the personal cost-benefit evaluation. In this sense, I would argue that it is (inseparable) the pandemic and governmental restrictions, and not only governmental restrictions (as concluded in line 233/234).

We fully agree with this point conceptually: it is certainly true that these utility losses do not necessarily apply to the population who would not opt for these measures without government restrictions (and in the absence of Covid-19). We have added a short note on this important point in the revised Discussion where we write:

“Estimating the extent to which these losses are due to government measures vs. the pandemic itself is difficult; if the entire population had opted for these measures without government intervention, the utility losses reported here would be primarily due to the pandemic, and the net impact of government restrictions would have been zero. On the other hand, if nobody had opted for these measures in the absence of government legislation, most of the observed disutility would due to legislation rather than the pandemic itself.”

Anecdotally we would argue that individual willingness to engage in any of these measures (even wearing masks) was very limited in Switzerland, suggesting that the extent to which these measures were used substantially exceeded the extent to which the average Swiss person would have relied on these measures if this had been a private choice. This is however purely speculative, and in our view not central to this paper, which simply captures the utility losses associated with the measures frequently applied by governments in the past 2.5 years.

4. Framing:

The discrete choice makes an effort to use a neutral setting, which would protect against my last point. However, I am doubtful that simply stating the introductory sentence “Imagine a world without COVID-19” (line 98) is strong enough to “force subjects to think about the restriction by itself” (line 96). First, the majority of subjects will have never experienced such restrictions before Covid, and the intensity of restrictions and the short time lag to the removal of restrictions will potentially tight them strongly to Covid. Second, the previous questions on the time trade-off lead to an implicit Covid-framing. Third, some of the elicited restrictions are clearly possible in a world without Covid (such as travel restrictions in totalitarian regimes), others like “wearing masks in public” seem highly unlikely in a world without infectious disease. Therefore, I am not surprised that there is no difference to previous study with Covid-framing. In summary, I would be more careful in claiming that it is only the governmental restrictions, but would rather see results as the sum of governmental and Covid-imposed restrictions.

This is a fair point. It is hard to think of many of these measures in abstract ways, and this may of course affect the choices made by individuals. We agree with the final point and summary and have added it to the Discussion as mentioned above.

5. Discussion:

I strongly disagree with the interpretation and comparison in line 236 to 241. First, I don’t think estimated quality-adjusted life years lost due to restrictions are comparable to life years lost due to higher mortality in an uncontained pandemic – see also my first point. Second, even if we would accept such a comparison, the death of 1% of the population would have severe consequences on the rest of the population. In this case, the benchmark is not the “normal life” anymore. Instead, we would have to do a similar choice experiment as for the restrictions, namely “normal life with functioning health sector” versus “life with collapse of health sector” or “life with old relatives“ versus “life with death of old relatives”. I don’t think it is far-fetched to assume that most people would not change 12 months of normal life against 12 months under such conditions.

We agree that this comparison was difficult and over-simplified things. We have revised this section as described above, and now more carefully distinguish between the impact of the pandemic and the impact of government restriction. As for the relative burden of disfunctioning health sectors, that is definitely an important aspect that is not easy to quantify (although we have tried to do so in a follow-up projects). In practice, many health systems were actually quite dysfunctional (especially for chronic care and for surgeries) for a long time with government measures, and it is not clear whether this state is strongly preferred to a counterfactual world with short waves of high mortality and stressed health systems.

As to emotional losses, that is of course always relevant, but it is not clear why a covid-19 death would have a higher emotional cost than deaths due to other causes.

This is a rather strong referee request, but I would ask the authors not to make this comparison between actual life year lost and quality adjusted-life years lost in the discussion and also not in Table 3 (see point 1). The comparison is simply not valid for methodological reasons and I think the paper makes an interesting contribution even without the comparison.

We have modified Table 3 as requested, and now show the robustness checks rather than the years of life lost due to the pandemic. We just mention the potential life years lost numbers briefly in the Discussion now, where we write:

“The overall disutility from life under restrictions is remarkably large. Under a rather pessimistic assumption that the pandemic would have resulted in the deaths of 1% of the total population the total number of life years lost would have been around 650,000. Our most conservative estimate for the total utility loss is close to 4 million QALYs. “

6. Willingness to pay:

Summing-up the average willingness to pay, subjects would be willing to spend on average a net monthly income of CHF 16,000 to avoid the restrictions. This amount is outside the monthly budget constraint of most individuals and seems highly inflated. The reason might be similar as before. Subjects who oppose the restriction might overstate their willingness to pay in order to express their strong opposition. The authors point the potentially overstated willingness to pay out in line 255 of the discussion but remain silent on potential reasons and implications.

My suggestion here would be to add that the elicited willingness to pay should not be taken at face value, and potentially to not state the numbers prominently in the abstract. Instead, I would recommend a relative interpretation of restrictions in comparison to the most (or least) “expensive” restrictions (e.g. on average, subjects are willing to pay 6.8-times as much to be allowed to do private parties than to not have to wear masks in public).

Thank you for this suggestion. We have added the following text to the Discussion as suggested:

“On average, respondents indicated to be willing to give up a bit more than CHF 600 per month for not having to wear masks, and more than CHF 4000 per month for not being allowed to have private parties or for not having to teach children at their homes. These estimates seem large, especially for individuals with income below the median and should be interpreted with caution, as it is not clear whether all subjects would really make these choices when faced with them in reality. Nevertheless, the estimates presented here suggest that most respondents would be willing to give up a substantial share of their income to avoid restrictions, and that restrictions on schooling, private parties and on going out are particularly undesired by the Swiss population.” 

Minor issues:

7. Explanation of approach: Complementary to the description how the authors calculated the quality-adjusted life years with the utility weights in the text, I think stating the formal estimation approach would ease the understanding (potentially accompanied by an example). This might be also information for an additional appendix.

Thank you for this suggestion. We now show all parameters underlying these calculations in a newly added appendix section.

8. Typos

- line 89: should it read “live” instead of “leave”?

- line 93: missing space “was asked”

- line 106: “each subject”

- line 119 to line 123: it should read “months” instead of “years”? (questions before are referring to “months”)

- line 163: it should read “normal life” instead of “health life”?

- line 172: it should be either “for” or “among”?

- line 198 (Table 2): some confidence intervals are not displayed correctly (minus sign and brackets covered)

Thank you for catching these – all of these typos have been fixed.

---

## [Decision Letter · Decision Letter 1]

12 Mar 2023

Subjective Burden of Government-imposed Covid-19 Restrictions in Switzerland:

Evidence from the 2022 LINK Covid-19 Survey

PONE-D-22-28984R1

Dear Dr. Fink,

We’re pleased to inform you that your manuscript has been judged scientifically suitable for publication and will be formally accepted for publication once it meets all outstanding technical requirements.

Kind regards,

Prof. Dr. Florian Follert

Academic Editor

PLOS ONE

Reviewers' comments:

Reviewer's Responses to Questions

**Comments to the Author**

1. If the authors have adequately addressed your comments raised in a previous round of review and you feel that this manuscript is now acceptable for publication, you may indicate that here to bypass the “Comments to the Author” section, enter your conflict of interest statement in the “Confidential to Editor” section, and submit your "Accept" recommendation.

Reviewer #2: All comments have been addressed

2. Is the manuscript technically sound, and do the data support the conclusions?

Reviewer #2: Yes

3. Has the statistical analysis been performed appropriately and rigorously? 

Reviewer #2: Yes

4. Have the authors made all data underlying the findings in their manuscript fully available?

Reviewer #2: Yes

5. Is the manuscript presented in an intelligible fashion and written in standard English?

Reviewer #2: Yes

6. Review Comments to the Author

Reviewer #2: Thank you very much for the opportunity to comment on the revised version of the paper. I feel the authors have incorporated my comments and suggestion very thoughtfully, and addressed all of my crucial and major points in a comprehensive way. This holds in particular for the requested robustness checks and the revised Table 3, which now allows the reader to compare different estimation scenarios. I’m happy to see their research published and think they make an important contribution to the debate on the effects of the Covid-19 restrictions.

Upon reading the revised version, there are two minor points that the authors might want to include, but I would leave it up to them and would suggest an acceptance of the final version without a further round with the reviewers.

1. In the instructions for participants (Appendix 1, second paragraph for each language), participants were told that the survey experiment was about Covid-19 and it’s impact on the daily life. This information might have made it particular attractive for people with strong opinions about Covid-19 restrictions to participate, and may explain the rather high fraction of people willing to trade-in zero or twelve months of their normal life. This is of course speculative, but might be mentioned in the discussion (line 272 following). Given that the results are robust to the exclusion of extreme preferences, it seems unlikely to me that such kind of selection drives the results – still, readers might be interested about explanations for the extreme preferences.

2. The third paragraph of the introduction mentions papers documenting welfare losses in different countries, but there is no paper on other German-speaking countries (which I would consider relevant). I’m aware of two studies in Germany that use the related concept of life satisfaction. Both find rather large effects of Covid-19 restrictions on life satisfaction in Germany, namely Konrad and Simon (2021)[ https://papers.ssrn.com/sol3/papers.cfm?abstract_id=3816728] and Bittmann (2022)[ https://link.springer.com/article/10.1007/s11482-021-09956-0]. I would suggest citing them as well.

7. PLOS authors have the option to publish the peer review history of their article (what does this mean?). If published, this will include your full peer review and any attached files.

Reviewer #2: No

---

## [Editor Report · Acceptance letter]

31 Mar 2023

PONE-D-22-28984R1 

Subjective Burden of Government-imposed Covid-19 Restrictions in Switzerland:
Evidence from the 2022 LINK Covid-19 Survey 

Dear Dr. Fink:

I'm pleased to inform you that your manuscript has been deemed suitable for publication in PLOS ONE. Congratulations! Your manuscript is now with our production department. 

Kind regards, 

on behalf of

Prof. Dr. Florian Follert 

Academic Editor

PLOS ONE